# Risk and protective factors for post-traumatic stress among New Zealand military personnel: A cross sectional study

Amy Richardson[1], Gagan Gurung[2], Ari Samaranayaka[3], Dianne Gardner[4], Brandon deGraaf[1], Emma H. Wyeth[5], Sarah Derrett[1], Daniel Shepherd[6], David McBride[1]*

**1** Injury Prevention Research Unit, Department of Preventive and Social Medicine, Dunedin School of Medicine, University of Otago, Dunedin, New Zealand, **2** Department of Preventive and Social Medicine, Dunedin School of Medicine, University of Otago, Dunedin, New Zealand, **3** Centre for Biostatistics, Division of Health Sciences, University of Otago, Dunedin, New Zealand, **4** School of Psychology, Massey University, Palmerston North, New Zealand, **5** Ngāi Tahu Māori Health Research Unit, Department of Preventive and Social Medicine, Dunedin School of Medicine, University of Otago, Dunedin, New Zealand, **6** Department of Psychology, Auckland University of Technology, Auckland, New Zealand

* david.mcbride@otago.ac.nz

**Data Availability Statement:** Data from this study is unsuitable for public deposition due to the privacy of participant data. Data are anonymised, but contain information on deployments (including

## Abstract

### Background

Post-traumatic stress (PTS) is prevalent among military personnel. Knowledge of the risk and protective factors associated with PTS in this population may assist with identifying personnel who would benefit from increased or targeted support.

### Aims

To examine factors associated with PTS among New Zealand military personnel.

### Methods

For this cross-sectional study, currently serving and retired military personnel were invited to complete a questionnaire. The questionnaire included a measure of PTS (the Military Post-traumatic Stress Disorder Checklist; PCL-M), where scores ≥30 indicate the experience of significant PTS symptoms and scores ≥45 indicate a presumptive clinical diagnosis of post-traumatic stress. Potential risk and protective factors associated with PTS were examined using logistic regression modelling.

### Results

1817 military personnel completed the questionnaire. PCL-M scores were ≥30 for 549 (30%) participants and ≥45 for 179 (10%) participants. Factors associated with higher PCL-M scores were trauma exposure, older age, male sex, and Māori ethnicity. Factors associated with lower PCL-M scores were greater length of service, psychological flexibility, and better quality sleep.

location and duration), which could lead to some participants being identified. Furthermore, the participant information sheet, as required by the Southern Health and Disability Ethics Committee specifically contains the statement that 'all study data would be kept strictly confidential to the research team.' Qualified researchers may apply for data access with the research team at veterans. health@otago.ac.nz and/or hdecs@moh.govt.nz.

**Funding:** Authors with funding: DI, AR, AS. Funders:Veterans Medical Research Trust Fund (No website), Lottery Health https://www. communitymatters.govt.nz/lottery-health-research/, The Royal New Zealand Returned and Services Association www.rsa.org.nz. The funders had no role in study design, data collection and analysis, decision to publish, or preparation of the manuscript.

**Competing interests:** The authors have declared that no competing interests exist.

## Conclusions

PTS was found to be prevalent among New Zealand military personnel. The experience of trauma was strongly associated with PTS. However, factors such as psychological flexibility (the ability to adapt to changes in circumstances) and good sleep were protective, suggesting that these factors could be key targets for interventions designed to reduce PTS among military personnel in New Zealand.

## Introduction

In New Zealand, the Defence Force has three primary personnel groups: the Regular Force, Reserve Forces, and Civilians (including those employed by the Defence Force and working overseas) [1]. These military personnel are responsible for contributing to the defence, security and wellbeing of the country. Research from other countries suggests that while many military personnel cope well with their roles [2], they are exposed to higher rates of both military and non-military trauma compared to the general population [3–5], and can be at greater risk of experiencing post-traumatic stress (PTS) [6, 7]. An elevated risk has been identified among some military personnel even during periods of low deployment activity [8–10].

Two critical events are commonly described in the military 'life course'–achievement of veteran status through operational deployment, and transition from military to civilian life. Operational deployment and witnessing atrocities has been associated with PTS [11, 12], while the period of transition from the military to civilian life has been found to confer an elevated risk of suicide, regardless of deployment history [13]. A lack of support during this period (including social and family support) serves to amplify suicide risk, and has been found to contribute to the experience of PTS [14].

While the two critical events described have been identified across diverse military samples [14], estimates of the prevalence of PTS vary dramatically both across and within countries. For example, the prevalence of PTS identified among deployed members of the United Kingdom Armed Forces is estimated to be 6% [15]. This is lower than estimates reported for deployed military personnel serving in the United States, Australia, and Canada, which range from 8–20% [16, 17]. Although differences are partially attributable to variation in sampling strategies, research methods, and diagnostic thresholds, differences in exposure to risk factors are also likely to play a role [16]. Research in veteran populations suggests that factors that can negatively affect adaptation to deployment and civilian transition include: female gender, ethnicity, high number of and longer duration deployments, prior adverse life events, pre-existing psychological disorders, trauma exposure, and alcohol misuse [14, 18–21]. Conversely, sleep [22, 23], social support [24, 25], and psychological flexibility (the ability to flexibly choose behaviour that is in line with personal goals and values) [26] are factors that have potential to protect against poor mental health outcomes, including PTS.

A significant number of New Zealand military personnel have been exposed to high levels of combat-related trauma and others have been deployed on peace-keeping missions [27]. Moral injury is one stressor associated with peace-keeping missions, defined as "perpetrating, failing to prevent, bearing witness to, or learning about acts that transgress deeply held moral beliefs and expectations" [28]. Other stressors include the restrictive rules of engagement [29], monotony and boredom, personnel encounter difficulties, and separation from family [30]. Pre-deployment and follow-up stages are also important to consider; these were the most

stressful periods, and had the greatest effect on mental and physical health, in a longitudinal study of 277 New Zealand military personnel deployed on peacekeeping duties [27].

While 10% of a community sample of New Zealand Vietnam War veterans were found to experience PTS [31], the prevalence of PTS among New Zealand military personnel more generally has not previously been reported. One reason for this is low response rates to post-deployment screens for PTS [32]. Research is also yet to identify the key risk and protective factors associated with PTS in this population. These factors may differ to those that have been identified in other military populations because of the significant contextual variation that exists across countries [16]. For example, the United States has a Veterans Health Administration that provides comprehensive and integrated healthcare specifically tailored to meet the unique needs of military personnel whereas such a system does not exist in New Zealand. Information on risk and protective factors is important to detect individuals who would benefit from targeted support following transition from the military, in order to reduce their risk of experiencing PTS. The aims of this cross-sectional study were to: 1) determine the prevalence of PTS (symptomology and presumptive clinical cases), and 2) identify protective and risk factors most strongly associated with PTS in a large sample of New Zealand military personnel.

## Materials and methods

### Sample

An online or paper questionnaire was available during June to December 2018 for completion by military personnel living in New Zealand. We defined 'military personnel' as any individuals who had served in the New Zealand Defence Force on active duty or in the reserves, regardless of whether they were still in service or had separated from the military. All military personnel residing in New Zealand were eligible to participate in the study. With respect to exclusion criteria, individuals who had not served in the Armed Forces were excluded, such as members of the national police force and Fire and Emergency New Zealand volunteers and staff. We attempted to recruit as many personnel as possible through an intensive recruitment campaign. According to the New Zealand Defence Force (NZDF), the number of currently serving personnel as of June 2018 was 14770 individuals. Of these, 9354 were regular personnel (2127 Navy, 4673 Army, 2554 Air Force), 2420 were reserves (480 Navy, 1685 Army, 255 Air Force), and 2996 were civilian personnel. Unfortunately the total number of living retired personnel at the time of this questionnaire study could not be determined. Since the end of World War II, in excess of 94,000 New Zealand Operational Service Medals (OSM, indicating operational veteran status) have been awarded but many of these may have been awarded posthumously.

### Procedure

Two different strategies were used to invite different types of personnel to take part. The NZDF hosts a secure Defence Intranet Exchange System (DIXS), and in mid-June, a message with an invitation to participate, and a link to the study web site, was sent on DIXS to the 3874 currently serving regular and reserve New Zealand Defence Force (NZDF) OSM holders. Retired military personnel were recruited through study posters distributed to the 43 local social clubs of the Royal New Zealand Returned and Services Association (RSA) identified by the RSA national office to be 'veteran active.' Paper questionnaires were also deposited at the RSAs along with instructions for how to return these to the research team. Announcements about the study were made on military social media pages, and both retired and currently serving personnel were invited to participate through an announcement on the No Duff Charitable

Trust website; No Duff is a registered charity committed to providing confidential support for military personnel and their families in New Zealand [33].

All study invitations (including emails, posters, and announcements) described the study as a survey designed to examine the health and wellbeing of current and former service personnel. Potential participants were informed that results from the survey would be used to help identify military personnel who might benefit from extra support. No mention of PTS was made on any of the study advertisements, minimising the potential for responder bias. Study posters and announcements included a link to a website from which interested personnel could access the online questionnaire. Advertisements of the study also advised all potential participants that they could request a paper version of the questionnaire from the research team if this was their personal preference.

Military personnel who visited the study website were required to enter their name and email address. This resulted in a personalised link being sent to their email address from which they could complete an online secure version of the questionnaire. Upon accessing this, participants were provided with an information sheet that documented the benefits and risks associated with the study as well as contact details for organisations that provide support for New Zealand military personnel. The information sheet noted that the questionnaire would take approximately 20 minutes to complete, that all study data would be kept strictly confidential to the research team, and that all participants would go into a prize draw to win a weekend holiday for two in New Zealand. Paper versions of the study information sheet and questionnaire were posted out to all personnel who contacted the research team indicating that this was their preference. These were posted with a return postage envelope. The study received approval from the Southern Health and Disability Ethics Committee of New Zealand (15/STH/40/AM02). We consulted with the Ngāi Tahu Research Consultation Committee in order to assess the importance of the project to Māori, New Zealand's indigenous population.

## Measures

The questionnaire included standardised measures of PTS (outcome) and six exposures.

Potentially adverse exposures examined included trauma, general distress, and hazardous drinking. Protective exposures included social support, sleep, and psychological flexibility.

Symptoms of PTS were assessed using the post-traumatic stress disorder (PTSD) checklist–military version (PCL-M). The PCL-M includes 17 items that ask about DSM-IV symptoms of PTS related to stressful military experiences, with response options ranging from 1 'Not at all' to 5 'Extremely' [34]. A total symptom severity score is calculated by summing responses to each option (range = 17–85). While scores of 30–35 indicate significant PTS symptomology and probable cases of PTSD, scores of ≥45 indicate a presumptive PTSD diagnosis [34]. In the present investigation, we present findings in relation to the two different cut-offs. This is because their utility is dependent on whether PTS scores are to be used for clinical objectives (such as to inform decisions about referral for clinical evaluation) or to estimate prevalence in a particular population [35]. The PCL-M has been identified as a widely used and well-validated measure in military populations [36].

The Brief Trauma Questionnaire (BTQ) was used to assess exposure to trauma. The BTQ consists of 10 items that screen for a range of different traumatic experiences [31]. For each item, participants are required to respond 'yes' or 'no' to indicate whether they have experienced the event and if so, whether they considered the event to present a threat to life or serious injury, and whether or not the event resulted in serious injury. Exposure to an event is scored as positive if a respondent says 'yes' to indicate life threat or serious injury from combat trauma, a serious car accident, a natural disaster, life-threatening illness, and physical or sexual

abuse, or to indicate exposure to violent death [37]. The BTQ is considered a reliable and valid measure to assess trauma exposure in defence force personnel [38].

Symptoms of distress were screened for using the General Health Questionnaire 12 (GHQ-12). This measure includes 12 items with a four-point response scale [39], which participants use to describe how they have been feeling over the past few weeks e.g. 'better than usual, same as usual, less than usual, much less than usual'. Using the Likert scoring method, items are summed to yield an overall total score (range = 0–36), with higher scores indicating greater distress [39]. The GHQ-12 is a reliable and valid measure [39] that has been used among military personnel in a number of different countries e.g. [10, 40].

The AUDIT-C is a 3-item measure that was used to identify potentially hazardous drinking [41]. Each item is answered using five response options, with possible total scores ranging from 0 to 12 [41]. A total score of ≥3 for women, and ≥4 for men, was used to identify participants engaging in hazardous drinking. The AUDIT-C has been validated in veteran populations [42, 43].

The Social Provisions Scale (SPS) was used to examine participant perceptions of the availability of different dimensions of social support. This theory-based social support instrument includes 24 items distributed across six subscales: reliable alliance, attachment, nurturance, social integration, reassurance of worth, and guidance [44]. Each item is rated on a 4-point Likert scale with responses ranging from 'strongly disagree' to 'strongly agree' [44]. After reversal of negatively worded items a total score was computed by summing all items (range = 24–96). Higher scores (including total scores and individual subscale scores) indicate higher levels of perceived social support. The construct validity, internal consistency, and test-retest reliability of the measure has been established across diverse populations [44, 45]. Scores on the SPS have been associated with psychological outcomes in military personnel [46, 47].

To screen for insomnia disorder based on DSM-5 criteria we used the 8-item Sleep Condition Indicator (SCI) [48]. All items were scored on a four point scale (0 to 4), with possible summed scores on this measure range from 0 to 32; scores were then re-scaled to a 0 to 10 scale [48]. Higher scores are indicative of better sleep. While the SCI is a relatively new measure, it has been validated in a number of different countries in a diverse range of populations [49].

To evaluate psychological flexibility, the 10-item Acceptance and Action Questionnaire II (AAQ-II) was used [50]. Responses to each item are made on a 7-point response scale, with options ranging from 'never true' to 'always true'. After reversing negatively worded items, the items of the scale were summed to obtain a total score (possible range 10 to 70), with higher scores indicative of greater psychological flexibility (less experiential avoidance) [50]. The AAQ-II has demonstrated internal construct validity in mild to moderately depressed and anxious populations [51], and scores on this measure have been identified as an important predictor of PTS among trauma-exposed military personnel [52].

The questionnaire also included a series of sociodemographic questions (gender, ethnicity, marital status, education level, employment status) and questions about service history, rank, and deployments.

## Analyses

Statistical analyses were completed using Stata 15 software [53]. First, descriptive statistics were used to describe the demographic characteristics of participants. Next, exploratory analyses investigated univariate associations between demographic, hazardous and protective factors, and the PTS outcomes (PCL-M ≥30 and PCL-M ≥45), with odds ratios (ORs) and 95% confidence intervals (CIs) estimated using logistic regression. Following this, multivariable

logistic regression (adjusted for age, sex, service years, and deployment status) was performed to identify exposures associated with PTS using a backward elimination process; variables with a *p*-value >0.10 were sequentially dropped from the model. With respect to missing data, if only one item was missing on a particular measure then this was imputed with the mean of the remaining items; if more than one item was missing then the participant was excluded from the analyses. The only exception to this was for the GHQ-12, where the standard procedure to count omitted items as low scores (0) was followed [39]. Only participants with complete data were included in the multivariable analyses.

In order to examine six risk and protective factors significantly associated with PTS (trauma exposure, distress, hazardous drinking, social support, sleep and psychological flexibility), a sample of at least 600 participants was required. These variables were chosen because they are potentially modifiable and have previously been associated with PTS in studies involving military personnel. We endeavoured to recruit as many participants as possible to maximise the generalisability of our findings.

## Results

A total of 1817 military personnel completed the questionnaire; 90 of the participants completed a paper version. Among participants, 549 (30%) reported PCL-M scores of ≥30 indicating significant PTS symptomology (probable PTSD), and 179 (10%) reported PCL-M scores of ≥45, indicative of presumptive clinical PTS.

The demographic characteristics of participants categorised according to the experience of probable (scores ≥30) and clinically significant PTS (scores ≥45) are presented in Table 1.

The median age of participants was 49.1 years (interquartile range, IQR = 38.7–61.1 years). The majority were male (87%) and were of New Zealand European ethnicity (78%); 14% of participants identified as Māori, similar to that of the NZDF as a whole, reported as 15%. Most participants had served in the military for at least 10 years (80%) and had been deployed at least once (84%). A majority of participants were currently serving (56%).

A median score of 11 was found for the 1735 participants who completed the GHQ-12 (IQR = 8–14; mean = 11.9; standard deviation, SD = 5.1), a median score of 75 was found for the 1778 participants who completed the SPS (IQR = 40–96; mean = 75.8; SD = 10.8), a median score of 54 was found for the 1734 participants who completed the AAQ-II (IQR = 46–60; mean = 52.3; SD = 10.1), and a median score of 5.6 was found for the 1711 participants who completed the SCI (IQR = 4.4–7.8; mean = 5.9, SD = 2.2).

Of the 1656 participants who completed the AUDIT-C, 898 (54%) reported hazardous drinking. Of the 1715 participants who completed the BTQ, the majority (*n* = 1187, 69%) had been exposed to trauma (see S1 Table). 1006 (59%) had served in a war zone and 736 (73%) of these individuals reported that this presented a threat to life and/or a threat of serious injury. The proportion of participants experiencing other traumatic events, including childhood physical and sexual abuse, was also high (35% and 16% respectively).

### Univariate analyses

Results of univariate analyses describing associations between exposure variables (demographic, risk, and protective factors) and PTS are presented in Table 2, showing both the odds of experiencing symptoms of PTS (PCL-M scores ≥30) and the odds of experiencing clinically relevant PTS (scores ≥45). With respect to PTS symptomology, older age, male sex, higher distress, and exposure to trauma were significantly associated with increased likelihood of PTS symptoms. In contrast, increased number of years in service, current participation in service, social support, psychological flexibility, and sleep were significantly associated with lower odds

**Table 1. Characteristics of participants according to PCL-M scores.**

| Characteristic | PCL-M Score ≥30 | | PCL-M Score ≥45 | | Total |
|---|---|---|---|---|---|
| | Low PCL-M Score 17–29 (*n* = 1268) | High PCL-M Score ≥30 (*n* = 549) | Low PCL-M Score 17–44 (*n* = 1638) | High PCL-M Score ≥45 (*n* = 179) | *n* = 1817 |
| *Age (years)* | | | | | |
| 20–29 | 124 (84%) | 24 (16%) | 141 (95%) | 7 (5%) | 148 (8%) |
| 30–39 | 264 (75%) | 86 (25%) | 335 (96%) | 15 (4%) | 350 (19%) |
| 40–49 | 327 (71%) | 134 (29%) | 418 (91%) | 43 (9%) | 461 (25%) |
| 50–59 | 247 (69%) | 111 (31%) | 331 (92%) | 27 (8%) | 358 (20%) |
| 60–69 | 176 (63%) | 103 (37%) | 240 (86%) | 39 (14%) | 279 (15%) |
| 70+ | 127 (59%) | 89 (41%) | 169 (78%) | 47 (22%) | 216 (12%) |
| missing | 3 (60%) | 2 (40%) | 4 (80%) | 1 (20%) | 5 (1%) |
| *Sex* | | | | | |
| Female | 183 (77%) | 54 (23%) | 224 (95%) | 13 (5%) | 237 (13%) |
| Male | 1065 (69%) | 488 (31%) | 1389 (89%) | 164 (11%) | 1553 (85%) |
| missing | 20 (74%) | 7 (26%) | 25 (93%) | 2 (7%) | 27 (2%) |
| *Ethnicity* | | | | | |
| NZ European | 997 (70%) | 418 (30%) | 1289 (91%) | 126 (9%) | 1415 (78%) |
| Māori | 177 (69%) | 79 (31%) | 218 (85%) | 38 (15%) | 256 (14%) |
| Other | 94 (64%) | 52 (36%) | 131 (90%) | 15 (10%) | 146 (8%) |
| *Service Years* | | | | | |
| 0–9 | 213 (62%) | 132 (38%) | 290 (84%) | 55 (16%) | 345 (19%) |
| 10–19 | 350 (70%) | 153 (30%) | 454 (90%) | 49 (10%) | 503 (28%) |
| 20–29 | 390 (73%) | 144 (27%) | 487 (91%) | 47 (9%) | 534 (29%) |
| 30–39 | 181 (69%) | 80 (31%) | 244 (93%) | 17 (7%) | 261 (14%) |
| 40–49 | 43 (65%) | 23 (35%) | 57 (86%) | 9 (14%) | 66 (4%) |
| missing | 91 (84%) | 17 (16%) | 106 (98%) | 2 (2%) | 108 (6%) |
| *Deployed* | | | | | |
| No | 186 (67%) | 92 (33%) | 245 (88%) | 33 (12%) | 278 (15%) |
| Yes | 1012 (71%) | 415 (29%) | 1295 (91%) | 132 (9%) | 1427 (79%) |
| missing | 70 (63%) | 42 (37%) | 98 (88%) | 14 (12%) | 112 (6%) |
| *Service Status* | | | | | |
| Retired | 453 (57%) | 337 (43%) | 650 (82%) | 140 (18%) | 790 (43%) |
| Currently Serving | 805 (80%) | 204 (20%) | 973 (96%) | 36 (4%) | 1009 (56%) |
| missing | 10 (55%) | 8 (45%) | 15 (83%) | 3 (17%) | 18 (1%) |
| *Hazardous Drinking* | | | | | |
| No | 537 (71%) | 221 (29%) | 677 (89%) | 81 (11%) | 758 (42%) |
| Yes | 625 (70%) | 273 (30%) | 819 (91%) | 79 (9%) | 898 (49%) |
| missing | 106 (66%) | 55 (34%) | 142 (88%) | 19 (12%) | 161 (9%) |
| *Trauma Exposure* | | | | | |
| No | 460 (87%) | 68 (13%) | 517 (98%) | 11 (2%) | 528 (29%) |
| Yes | 743 (63%) | 444 (37%) | 1031 (87%) | 156 (13%) | 1187 (65%) |
| missing | 65 (64%) | 37 (36%) | 90 (88%) | 12 (12%) | 102 (6%) |

**Table 2. Univariate associations between exposure variables and elevated PCL-M scores (≥30 and ≥45 respectively).**

| Characteristic | PCL-M Score ≥30 | | | | | | | PCL-M Score ≥45 | | | | | | |
|---|---|---|---|---|---|---|---|---|---|---|---|---|---|---|
| | N PCL-M Score <30 | N PCL-M Score ≥30 | AR | OR | 95% CI for OR | p | n | N PCL-M Score <45 | N PCL-M Score ≥45 | AR | OR | 95% CI for OR | p | n |
| *Age (Years)** | * | * | | 1.02 | 1.01, 1.03 | <0.01 | 1812 | * | * | | 1.03 | 1.02, 1.05 | <0.01 | 1812 |
| *Sex* | | | | | | | | | | | | | | |
| Female | 183 | 54 | | Ref | | | | 224 | 13 | | Ref | | | |
| Male | 1065 | 488 | 0.09 | 1.55 | 1.13, 2.14 | 0.01 | 1790 | 1389 | 164 | 0.05 | 2.03 | 1.14, 3.64 | 0.02 | 1790 |
| *Ethnicity* | | | | | | | | | | | | | | |
| NZ European | 997 | 418 | | Ref | | | | 1289 | 126 | | Ref | | | |
| Māori | 177 | 79 | 0.01 | 1.06 | 0.80, 1.42 | 0.67 | | 218 | 38 | 0.06 | 1.78 | 1.21, 2.63 | <0.01 | |
| Other | 94 | 52 | 0.06 | 1.32 | 0.92, 1.89 | 0.13 | 1817 | 131 | 15 | 0.01 | 1.17 | 0.67, 2.06 | 0.58 | 1817 |
| *Service Years** | * | * | | 0.99 | 0.98, 1.00 | 0.03 | 1709 | * | * | | 0.97 | 0.96, 0.99 | <0.01 | 1709 |
| *Deployed* | | | | | | | | | | | | | | |
| No | 186 | 92 | | Ref | | | | 245 | 33 | | Ref | | | |
| Yes | 1012 | 415 | -0.04 | 0.83 | 0.63, 1.10 | 0.18 | 1705 | 1295 | 132 | -0.03 | 0.76 | 0.50, 1.13 | 0.18 | 1705 |
| *Currently Serving* | | | | | | | | | | | | | | |
| No | 453 | 337 | | Ref | | | | 650 | 140 | | Ref | | | |
| Yes | 805 | 204 | -0.22 | 0.34 | 0.28, 0.42 | <0.01 | 1799 | 973 | 36 | -0.14 | 0.17 | 0.12, 0.25 | <0.01 | 1799 |
| *Distress** | * | * | | 1.27 | 1.23, 1.30 | <0.01 | 1735 | * | * | | 1.21 | 1.18, 1.25 | <0.01 | 1735 |
| *Social Support** | * | * | | 0.91 | 0.90, 0.93 | <0.01 | 1778 | * | * | | 0.91 | 0.90, 0.93 | <0.01 | 1778 |
| *Psychological Flexibility** | * | * | | 0.84 | 0.82, 0.85 | <0.01 | 1734 | * | * | | 0.84 | 0.82, 0.86 | <0.01 | 1734 |
| *Sleep** | * | * | | 0.53 | 0.49, 0.57 | <0.01 | 1711 | * | * | | 0.41 | 0.36, 0.46 | <0.01 | 1711 |
| *Hazardous Drinking* | | | | | | | | | | | | | | |
| No | 537 | 221 | | Ref | | | | 677 | 81 | | Ref | | | |
| Yes | 625 | 273 | 0.01 | 1.06 | 0.86, 1.31 | 0.58 | 1656 | 819 | 79 | -0.02 | 0.81 | 0.58, 1.12 | 0.20 | 1656 |
| *Trauma Exposure* | | | | | | | | | | | | | | |
| No | 460 | 68 | | Ref | | | | 517 | 11 | | Ref | | | |
| Yes | 743 | 444 | 0.25 | 4.04 | 3.05, 5.35 | <0.01 | 1715 | 1031 | 156 | 0.11 | 7.11 | 3.82, 13.23 | <0.01 | 1715 |

*Continuous variable (no reference group); AR = absolute risk.

of experiencing PTS symptoms. The same pattern of associations was also found for clinically relevant PTS, in addition to greater odds of clinical PTS among individuals identifying as Māori compared to those of NZ European ethnicity.

## Multivariate analyses

Results of multivariate analyses describing associations between exposure variables and PTS, after adjustment for age, sex, service years and deployment status, are presented in Table 3, including for odds of experiencing PTS symptomology (PCL-M scores ≥30) and for odds of clinically relevant PTS (scores ≥45). With respect to PTS symptomology, older age, male sex, higher distress, and exposure to trauma were significantly associated with an increased likelihood of PTS. Increased number of years in service, psychological flexibility, and sleep were significantly associated with decreased odds of experiencing PTS symptoms; social support was no longer significantly associated with this outcome. A single unit increase in sleep score corresponded to a 30% reduction in odds of experiencing significant PTS symptoms. The same pattern of results was found for clinically-relevant PTS, with the exception of higher distress,

**Table 3. Multivariate associations between exposure variables and elevated PTSD (scores ≥ 30 and scores ≥ 45).**

| Characteristic | PCL-M Score ≥30, n = 1532 | | | | | PCL-M Score ≥45, n = 1567 | | | | |
|---|---|---|---|---|---|---|---|---|---|---|
| | N PCL-M Score <30 | N PCL-M Score ≥30 | OR | 95% CI for OR | p | N PCL-M Score <45 | N PCL-M Score ≥45 | OR | 95% CI for OR | p |
| *Age (Years)** | * | * | 1.02 | 1.01, 1.03 | <0.01 | * | * | 1.04 | 1.03, 1.06 | <0.01 |
| *Sex* | | | | | | | | | | |
| Female | 145 | 45 | Ref | | | 152 | 46 | Ref | | |
| Male | 915 | 427 | 1.84 | 1.14, 2.98 | 0.01 | 933 | 436 | 1.69 | 0.74, 3.86 | 0.21 |
| *Ethnicity* | * | * | | | | | | | | |
| NZ European | | | | | | 856 | 367 | Ref | | |
| Māori | | | | | | 151 | 71 | 2.80 | 1.54, 5.10 | <0.01 |
| Other | | | | | | 78 | 44 | 0.97 | 0.40, 2.31 | 0.94 |
| *Service Years** | * | * | 0.98 | 0.97, 1.00 | 0.01 | * | * | 0.97 | 0.95, 0.99 | <0.01 |
| *Deployed* | | | | | | | | | | |
| No | 162 | 83 | Ref | | | 166 | 84 | Ref | | |
| Yes | 898 | 389 | 1.31 | 0.85, 2.00 | 0.22 | 919 | 398 | 1.54 | 0.84, 2.81 | 0.16 |
| *Distress** | * | * | 1.07 | 1.03, 1.11 | <0.01 | | | | | |
| *Psychological Flexibility** | * | * | 0.87 | 0.85, 0.89 | <0.01 | * | * | 0.87 | 0.85, 0.90 | <0.01 |
| *Sleep** | * | * | 0.70 | 0.64, 0.77 | <0.01 | * | * | 0.56 | 0.49, 0.66 | <0.01 |
| *Hazardous Drinking* | | | | | | | | | | |
| No | 500 | 212 | Ref | | | | | | | |
| Yes | 560 | 261 | 1.11 | 0.96, 1.77 | 0.08 | | | | | |
| *Trauma Exposure* | | | | | | | | | | |
| No | 396 | 61 | Ref | | | 405 | 62 | Ref | | |
| Yes | 664 | 411 | 3.03 | 2.07, 4.41 | <0.01 | 680 | 420 | 3.34 | 1.54, 7.27 | <0.01 |

Variables with a *p*-value less than 0.10 after adjustment for age, sex, service years, and deployment status are included in the model.

*Continuous variable (no reference group).

which was not significantly associated. In addition, Māori participants were found to have greater odds of experiencing clinically relevant PTS when compared to New Zealand Europeans.

## Discussion

This cross-sectional study identified a high prevalence of PTS in a large sample of currently serving and retired New Zealand military personnel. Thirty percent of participants reported experiencing probable PTS and 10% were identified as having clinically relevant PTS. These findings are similar to those reported in an earlier study, which found evidence of PTS among 10% of New Zealand Vietnam war veterans [31]. Our results indicate that it is not only retired personnel who are at risk. While substantial differences in PTS have been documented in response to different traumatic events [54], the weighted prevalence of PTS in response to war related trauma (3.5%) is similar to the weighted prevalence of PTS associated with random traumatic events (4%) [55]. This may explain why the prevalence of probable PTS found in our study is similar to that documented in other trauma population samples [35]. Our results highlight that support to deal with PTS is needed for a large number of New Zealanders who are serving, or have served, in the military.

As a consequence of the sampling methods used and the limited response rate to our survey, findings can provide only a rough estimate of the prevalence of PTS. Nevertheless, they

suggest that the prevalence of clinically significant PTS is higher among military personnel compared with the general population of New Zealand, where the prevalence is estimated to be 3% [56]. The finding that 10% of participants reported symptoms indicative of a clinical diagnosis of PTSD is in line with point prevalence estimates of combat-related PTSD in US military veterans, ranging from approximately 2% to 17% [6]. Summary estimates of PTSD prevalence for military personnel and veterans from a number of countries range from 1.1% to 34.8% [14]. The prevalence of PTS identified in our study is higher than that documented among UK military personnel [15], although this may be attributable to variation in sampling strategies and data collection methods used. Participants in the UK study were individually invited to participate in an online survey via postal invitation, with non-responders sent a repeat invitation and subsequent postal questionnaire [15]. In addition, there was an intensive period of follow-up and tracing of non-responders. Differences in PTS estimates may also be due to variation in socio-political and cultural factors that vary across nations [6].

Trauma exposure was most strongly associated with odds of experiencing PTS symptomology and clinically relevant PTS in the present study, and is a prerequisite for a DSM IV diagnosis of PTSD. General distress was significantly associated with increased odds of PTS symptoms, although was not significantly associated with odds of clinical PTS after adjustment for age, sex, service years, and deployment status. This is consistent with findings from a meta-analysis of risk factors for combat-related PTS among military personnel and veterans, which did not identify general distress to be a significant risk factor, although a history of prior psychological problems and trauma exposure were [14].

Māori participants had greater odds of reporting clinical PTS than their New Zealand European counterparts. Higher levels of PTS among Māori were also detected in a sample of 756 New Zealand Vietnam War veterans, however, the effect of ethnicity on PTS was mediated by higher levels of combat stressors experienced by Māori, including stressors related to combat exposure, rank, and combat role [57].

Consistent with findings of a study examining predictors of persistent PTS in UK military personnel [58], older age was significantly associated with increased odds of experiencing PTS. However, in contrast to other studies examining PTS in military personnel, males were at greater risk of experiencing PTS than females. It is important to note that other studies identifying females to be at greater risk of PTS have focused on combat-related PTS [14], and our sample includes personnel who never deployed. A 2014 NZDF report noted that only 6% of officers in combat/operations were women [59]. Therefore, greater exposure to combat-related trauma among male military personnel in New Zealand may explain why they reported more PTS symptoms than female personnel.

Despite older age being associated with increased odds of PTS, a greater number of service years was associated with reduced odds of PTS. Rather than this reflecting a causal role of service duration in relation to PTS, this may reflect a resilience that develops over time among long-serving personnel, or may be due to individuals with PTS leaving military service earlier as has been reported among UK Armed Forces Personnel [60].

Better self-reported sleep was also associated with fewer PTS symptoms among military personnel in our sample. Sleep-related problems have been identified as the most frequently endorsed PTS symptom, occurring in 60–90% of people with the disorder [61]. Sleep disturbance is highly prevalent among military personnel. In a sample of 80 personnel who had recently returned from combat, 74% reported insomnia and 61% experienced distressing nightmares [62]. Sleep disturbance was associated with higher PTS and depression severity scores and these associations persisted over a six month period [62]. While PTS and depression decreased in the sample over time, insomnia remained prevalent. Therefore, the bidirectional relationship between sleep disturbance and PTS may be perpetuated by poor sleep

exacerbating other symptoms of PTS and diminishing the capacity of military personnel to manage these.

There is preliminary evidence to indicate that early intervention among military personnel experiencing sleep disturbance may help to reduce PTS symptoms. An investigation of 44 military personnel who received cognitive behavioural therapy for insomnia found that participants who experienced improved sleep quality from pre- to post-treatment (n = 28) had significant declines in depression and PTS symptoms [23]. In contrast, those whose sleep did not improve had no changes in their psychiatric symptoms, as well as a reduction in health-related quality of life.

In the present study, psychological flexibility was associated with reduced odds of reporting PTS. However, it is important to interpret our findings with caution as there have been criticisms of the AAQ-II, with several researchers arguing that it may be measuring psychological distress rather than psychological inflexibility [63]. Furthermore, numerous versions of the AAQ-II exist which can make it difficult to compare findings across studies. Despite few studies examining psychological flexibility among military personnel, techniques designed to increase psychological flexibility (such as acceptance and commitment therapy) are being investigated as potential treatments for PTS in this population [64–66], and evidence for their effectiveness is emerging [67].

Strengths of this study include the large sample, and the inclusion of those who have never been deployed. Our study also serves to provide a snapshot of New Zealand military personnel, for which the total number of those who have served and are alive is unknown. We do however know that the invitation was specifically sent to 3874 NZDF members who were 'currently serving' veterans with access to the military system in June 2018, and that the majority of responses were received in the following few weeks. Although the number of individuals in the sample who had never been deployed was small, our findings suggest that these individuals are also at risk of PTS. Evidence that personnel who have never deployed are at greater risk of PTS than the general population is growing [10, 68]. It is clear that factors other than deployment have an important role to play in the experience of PTS among military personnel.

It is also important to acknowledge the limitations associated with this study. At the time the questionnaire was distributed there were 14770 currently serving personnel in the NZDF, of whom 9354 were regular personnel, 2420 were reservists and 2996 were civilians. Very few reservists will have received the email invitation to participate: access to the secure email system through which the link to the study was distributed is extremely limited unless in a 'command' role. The majority of questionnaires were completed by currently serving personnel, indicating a response rate in the order of 19% when excluding both reservists and civilians. This response rate and the large proportion of currently serving personnel in our sample gives rise to potential sampling bias. Results may not be generalisable to all New Zealand military personnel, particularly those who are no longer serving. It is also important to note that military personnel with higher PTS may have been more likely to participate, giving rise to inflated estimates of PTS prevalence. Conversely, our findings may underestimate the prevalence of PTS in this population if those without PTS were more inclined to participate.

The response rate to this study was low relative to other studies conducted with military personnel which is likely attributable to the recruitment method used. In contrast to studies that recruited directly from veteran-specific treatment programs [69, 70], we did not approach participants directly. Other studies have targeted pre-defined populations (e.g. US Gulf veterans) for which information on the approximate number of personnel is available, in addition to opportunities for direct contact e.g. [71]. It is possible that New Zealand military personnel were not exposed to study advertisements and were therefore unaware that the study was

taking place; this is particularly true of retired military personnel who were not emailed about the study.

Information on the way in which participants were recruited (for example, from an email invitation or alternative study advertisement) was not collected. This precluded sensitivity analyses comparing estimates with respect to the different sampling strategies used. With respect to missing data, complete case analyses was used as the assumptions necessary for multiple imputation were not met by the missing data within our sample. Another limitation of the study is the cross-sectional design which precludes the identification of cause and effect relationships between exposures and PTS. Additionally, although we assessed and accounted for a range of known confounders, it is possible some important confounders were not considered and that these may explain significant relationships between exposures and PTS.

It is unclear how generalisable the study findings are to countries outside of New Zealand, where the characteristics of military personnel, deployment experiences, and post-deployment support services are likely to differ [16]. Nevertheless, health support for military personnel in New Zealand follows that provided by American, Canadian, British, and Australian Defence Forces.

## Conclusions

Knowledge of the risk and protective factors associated with PTS may allow for early identification of military personnel who would benefit from targeted support to promote their wellbeing. Our findings suggest that trauma exposure is most strongly associated with high levels of PTS, while good sleep and psychological flexibility are protective. As these protective factors are amenable to standardised measurement and modification, screening could facilitate early intervention to prevent or reduce PTS. Future research is needed to identify whether relationships in our study can be found longitudinally. This would establish the sequence of events (for example, whether changes in sleep are associated with subsequent changes in PTS), providing further evidence regarding New Zealand military personnel most at risk of experiencing PTS.

## Supporting information

**S1 Table. Participant responses to the Brief Trauma Questionnaire (BTQ), (N = 1817).** (DOCX)

## Acknowledgments

The authors would like to thank all New Zealand military personnel who participated in this study.

## Author Contributions

**Conceptualization:** Amy Richardson, Ari Samaranayaka, Dianne Gardner, Emma H. Wyeth, Sarah Derrett, Daniel Shepherd, David McBride.

**Data curation:** Gagan Gurung, Brandon deGraaf.

**Formal analysis:** Ari Samaranayaka.

**Funding acquisition:** Amy Richardson, Emma H. Wyeth, Sarah Derrett, Daniel Shepherd, David McBride.

**Investigation:** Amy Richardson, Gagan Gurung, Brandon deGraaf, Emma H. Wyeth, Sarah Derrett, Daniel Shepherd, David McBride.

**Methodology:** Amy Richardson, Ari Samaranayaka, Dianne Gardner, Emma H. Wyeth, Sarah Derrett, David McBride.

**Project administration:** Amy Richardson, Gagan Gurung, Daniel Shepherd.

**Resources:** Brandon deGraaf.

**Software:** Brandon deGraaf.

**Supervision:** Ari Samaranayaka.

**Validation:** Gagan Gurung.

**Visualization:** Gagan Gurung.

**Writing – original draft:** Dianne Gardner, David McBride.

**Writing – review & editing:** Dianne Gardner, Brandon deGraaf, Emma H. Wyeth, Daniel Shepherd, David McBride.

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
