## [Decision Letter · Decision Letter 0]

22 Oct 2019

PONE-D-19-23666

Post-Traumatic Stress among New Zealand military personnel: a cross sectional study

PLOS ONE

Dear Dr. McBride,

Thank you for submitting your manuscript to PLOS ONE. After careful consideration, we feel that it has merit but does not fully meet PLOS ONE’s publication criteria as it currently stands. Therefore, we invite you to submit a revised version of the manuscript that addresses the points raised during the review process.

You will find that both reviewers have provided very helpful comments to help you improve the clarity and replicability of your study. I have also marked up some suggestions on a PDF of your submission, which I attach to this decision letter. It is especially important that you consider the combined feedback when providing further details on the study methods, the design of the analyses, and expansion of the discussion of some the key findings. Given that this is a cross-sectional study design it is important that assumptions of causality are not explicitly or implicitly made. 

We would appreciate receiving your revised manuscript by Dec 06 2019 11:59PM. To enhance the reproducibility of your results, we recommend that if applicable you deposit your laboratory protocols in protocols.io, where a protocol can be assigned its own identifier (DOI) such that it can be cited independently in the future. For instructions see: http://journals.plos.org/plosone/s/submission-guidelines#loc-laboratory-protocols

A rebuttal letter that responds to each point raised by the academic editor and reviewers. Please indicate the respective location in the track-edited version of the resubmitted manuscript where each point has been addressed. This letter should be uploaded as separate file and labeled 'Response to Reviewers'.A marked-up copy of your manuscript that highlights changes made to the original version. This file should be uploaded as separate file and labeled 'Revised Manuscript with Track Changes'.An unmarked version of your revised paper without tracked changes. This file should be uploaded as separate file and labeled 'Manuscript'.

We look forward to receiving your revised manuscript.

Kind regards,

Melita J. Giummarra

Academic Editor

PLOS ONE

Journal Requirements:

2.  Please include additional information regarding the survey or questionnaire used in the study and ensure that you have provided sufficient details that others could replicate the analyses. For instance, if you developed a questionnaire as part of this study and it is not under a copyright more restrictive than CC-BY, please include a copy, in both the original language and English, as Supporting Information."

Additional Editor Comments (if provided):

Reviewers' comments:

Reviewer's Responses to Questions

**Comments to the Author**

1. Is the manuscript technically sound, and do the data support the conclusions?

Reviewer #1: Yes

Reviewer #2: Partly

2. Has the statistical analysis been performed appropriately and rigorously? 

Reviewer #1: Yes

Reviewer #2: I Don't Know

3. Have the authors made all data underlying the findings in their manuscript fully available?

Reviewer #1: No

Reviewer #2: No

4. Is the manuscript presented in an intelligible fashion and written in standard English?

Reviewer #1: Yes

Reviewer #2: Yes

5. Review Comments to the Author

Reviewer #1: 1. The original sample and response rates are unclear. I suggest including a breakdown of how many people were invited to take part in the study. If people were not invited directly it should be made clear how the survey was made available or accessed (e.g. hosted on a website). While the number of people in the study is perhaps reasonable (1817) it is difficult to ascertain the response rate. Please provide the number of currently serving personnel, the number of current reserves, and the number of retired personnel. This way you can give good estimates of the response rate by group. e.g. are retired personnel underrepresented? Also in Table 1, provide information on the number of currently serving, reserve and retired.

2. Table 2 is not mentioned in the text and so currently serves little purpose. Perhaps these statistics could be mentioned in the text instead (and the table removed).

3. Table 4 is confusing because a number of variables with a p-value >0.1 are included in the table which seems to contradict the footnote (e.g. deployed, Female in the >=45 column).

4. Discussion. Why were reservists unlikely to receive the email invitation? More information about the recruitment methods are required to increase reader confidence in how the study was undertaken.

Reviewer #2: This paper explores the prevalence of and risk factors associated with post traumatic stress (PTS) in the New Zealand military. The survey was based on 1817 military personnel and reports a prevalence of PTS of 10%-30% depending on the cut-off used. The authors report on the factors associated with PTS and suggest factors that could be targeted to reduce PTS within military personnel.

The health and wellbeing of military and veteran personnel is receiving much attention internationally and this is one of the first studies emerging from New Zealand.

My main concern is regarding the methodology employed, this needs to be more clearly articulated in the paper. For example, further details of the size of the population approached (number of serving personnel, reserves and veterans), recruitment methods used and exclusion criteria applied are required. What attempts were made to ensure all currently serving and veteran personnel were given the opportunity to participate? This level of detail is needed to enable to reader to determine the possible presence of response bias and the generalisability of the results. Further details are required in the methods and the subsequent limitations and implications more clearly addressed in the discussion.

Introduction:

- the authors state that military personnel are at greater risk of PTS than the general populatlion. This is not the case in all nations and this should be recognised by the authors.

- please define the term veteran and be conscious that term is used differently across nations and studies.

- a limited number of references are provided to support the statements made, please ensure a broad range of references are included (where appropriate).

Procedure:

- this details the approach used for currently serving personnel, what about veterans?

- please clarify the sample size included and what is meant by cases? Should this be participants? If it is cases then the study is underpowered as less than 600 cases were identified. What are the implications of this for the analyses conducted and conclusions drawn?

Measures:

- need to consistently report details of all measures used, for example, range, response options and validity within the population under study.

Analyses:

- the authors state that missing data were dealt with using "developer recommendations", details need to be provided of what these recommendation are for each measure.

- only participants with complete data were included in the multivariable analysis, please state number and % of total sample.

- further justification for using two cut-offs for the PCL need to be provided. Why are two cut-offs being used? Could the authors focus on one cut-off for the paper with the other results being provided as supplementary material?

- has the analytical approach applied been discussed with a statistican?

Results:

- given the existing interest in differences between serving and ex-service personnel, all analyses should take serving status into account.

Table 1:

- need to report the overall numbers and %'s for each characteristic

- why are row and not column %'s reported?

Table 3:

- need to include numbers and %'s for each characteristic by PCL score (including those in the baseline category)

- a number of variables appear in this table which should be included in Tables 1 or 2, for example, hazardous drinking and trauma exposure

Discussion:

- how do the prevalence estimates reported within the military population compare with the general population of New Zealand?

Conclusion:

- please justify the statement regarding longitudinal data and how that will help identify those most at risk of PTS

Data availability:

- the authors state that the full data won't be made available and that some restrictions will apply. Could they clarify what will be made available and what restrictions will be in place?

6. PLOS authors have the option to publish the peer review history of their article (what does this mean?). If published, this will include your full peer review and any attached files.

Reviewer #1: Yes: Dr Michael Waller

Reviewer #2: No

---

## [Author Response · Author response to Decision Letter 0]

19 Dec 2019

Response

We would like to thank the reviewers and academic editor for their considered comments on our manuscript. These comments are addressed below. Changes to the manuscript have been made using track changes.

Reviewer One Comments:

1. The original sample and response rates are unclear. I suggest including a breakdown of how many people were invited to take part in the study. If people were not invited directly it should be made clear how the survey was made available or accessed (e.g. hosted on a website). While the number of people in the study is perhaps reasonable (1817) it is difficult to ascertain the response rate. Please provide the number of currently serving personnel, the number of current reserves, and the number of retired personnel. This way you can give good estimates of the response rate by group. e.g. are retired personnel underrepresented? Also in Table 1, provide information on the number of currently serving, reserve and retired.

A number of different approaches were used to invite people to take part in the study. However, the strategies used differed for currently serving and retired personnel. These strategies have now been clarified in the methods section of the manuscript (see page 5, paragraph 3; page 6, paragraph 1 and 3). 

An invitation and link to the online questionnaire were emailed to all currently serving military personnel and reservists in mid-June 2018. However, a different approach was used to invite retired personnel who were recruited through study posters distributed to the largest branches of the Royal New Zealand Returned and Services Association (RSA) and through announcements on social media. The posters and announcements contained a link to a study website where individuals could enter their name and email address. This resulted in a personalised link being sent to their email address from which they could complete an online secure version of the questionnaire. 

We have now described the number of currently serving and the number of retired personnel in our sample (see page 12, paragraph 1), as well as best possible estimates of the number of these personnel in the New Zealand population (see page 5, paragraph 2). Information on the number of currently serving personnel as of June 2018 was provided by the New Zealand Defence Force (NZDF). However, it is unclear how many living retired personnel there were in New Zealand at the time of our survey. Since the end of World War II, 76,000 operational service medals have been awarded but many of these may have been awarded posthumously. 

The number of currently serving personnel and the number of retired personnel who participated in the study is now reported in Table 1. 

2. Table 2 is not mentioned in the text and so currently serves little purpose. Perhaps these statistics could be mentioned in the text instead (and the table removed).

The results presented in Table 2 are now reported in text (see page 12, paragraph 2).

3. Table 4 is confusing because a number of variables with a p-value >0.1 are included in the table which seems to contradict the footnote (e.g. deployed, Female in the >=45 column).

The footnote notes that the analysis has been adjusted for age, sex, service years, and deployment status. This means that these 4 variables were retained in the model irrespective of their p-values. 

4. Discussion. Why were reservists unlikely to receive the email invitation? More information about the recruitment methods are required to increase reader confidence in how the study was undertaken.

Very few reservists will have received the email invitation as access to the secure email system through which the questionnaire was distributed is extremely limited unless in a ‘command’ role. The majority of reservists cannot access the secure system. This is now noted in the discussion (see page 21, paragraph 3) and more information about recruitment methods is now provided in the methods section of the manuscript (see page 5, paragraph 3; page 6, paragraph 1 and 3).

Reviewer Two Comments:

My main concern is regarding the methodology employed, this needs to be more clearly articulated in the paper. For example, further details of the size of the population approached (number of serving personnel, reserves and veterans), recruitment methods used and exclusion criteria applied are required. What attempts were made to ensure all currently serving and veteran personnel were given the opportunity to participate? This level of detail is needed to enable to reader to determine the possible presence of response bias and the generalisability of the results. Further details are required in the methods and the subsequent limitations and implications more clearly addressed in the discussion.

Please see response to reviewer one, comment one. 

With respect to inclusion criteria, all military personnel residing in New Zealand were eligible to participate in the study. With respect to exclusion criteria, individuals who had not served in the Armed Forces were excluded, such as members of the national police force and Fire and Emergency New Zealand volunteers and staff. This information has been added to the methods section (see page 5, paragraph 2). 

The limitations associated with our study design, and the implications of this for the generalisability of findings, have now been elaborated on in the discussion of the manuscript (see page 21, paragraph 3; page 22, paragraph 1, 2, 3). 

Introduction:

- The authors state that military personnel are at greater risk of PTS than the general population. This is not the case in all nations and this should be recognised by the authors.

We thank the reviewer for bringing this to our attention. This statement has been modified to acknowledge that military personnel can be at greater risk of PTS (see page 3, paragraph 1). 

- Please define the term veteran and be conscious that term is used differently across nations and studies.

Given the variation in the term veteran across nations and studies we endeavoured to consistently use the slightly broader term ‘military personnel’ throughout our manuscript. The different types of military personnel in New Zealand are described at the beginning of the introduction (see page 3, paragraph 1). We have also added a description of how we defined military personnel in our study to the beginning of the methods section (see page 5, paragraph 2). 

- A limited number of references are provided to support the statements made, please ensure a broad range of references are included (where appropriate).

Additional references have now been added to the introduction section. 

Procedure:

- This details the approach used for currently serving personnel, what about veterans?

Please see response to Reviewer One, Comment One. 

- Please clarify the sample size included and what is meant by cases? Should this be participants? If it is cases then the study is underpowered as less than 600 cases were identified. What are the implications of this for the analyses conducted and conclusions drawn?

We apologise for the confusion caused by the statement about cases. This statement was intended to identify the minimum number of participants that would be required to examine six risk and protective factors in multivariable models. The sentence has been rewritten and moved to the analyses section of the methods (see page 10, paragraph 2). The total number of participants included in the multivariable analyses is presented in the top row of Table 4 (1532 participants when examining factors associated with scores ≥30 and 1567 when examining factors associated with scores ≥45).

Measures:

- Need to consistently report details of all measures used, for example, range, response options and validity within the population under study.

An effort has been made to ensure that details of all measures are now reported consistently, including the response options for each measure, the possible range of total scores, and the validity of the measure within military personnel (see page 7, paragraph 4; page 8, paragraph 1, 2; page 9, paragraph 1, 2, 3). 

Analyses:

- The authors state that missing data were dealt with using "developer recommendations", details need to be provided of what these recommendation are for each measure.

With respect to missing data, if only one item was missing on a particular measure then this was imputed with the mean of the remaining items; if more than one item was missing then the participant was excluded from the analyses. The only exception to this was for the GHQ-12, where the standard procedure to count omitted items as low scores (0) was followed. This information is now reported in the analyses section (see page 10, paragraph 1). 

- Only participants with complete data were included in the multivariable analysis, please state number and % of total sample.

The number of participants included in the multivariable analyses is presented at the top of Table 4 (n = 1532 for analyses exploring PCL-M Score ≥30 and n = 1567 for analyses exploring PCL-M Score ≥45). Numbers can be used by readers to calculate the percentage of the total sample (84% and 86% respectively).

- Further justification for using two cut-offs for the PCL need to be provided. Why are two cut-offs being used? Could the authors focus on one cut-off for the paper with the other results being provided as supplementary material?

The two cut-offs were used to provide readers with information on the proportion of participants reporting PTS symptomology and the proportion of participants likely to meet clinical criteria for a PTSD diagnosis. Both are important to know about as the different cut-offs can be used for different purposes (e.g. clinical rather than population objectives). This, in turn, will alter the way in which findings are considered. In populations with high PTS prevalence (such as trauma-exposed populations), cut-off values of 44 or higher will likely underestimate prevalence. Therefore, if an aim is to use the PCL-M as a clinical detection tool, a low cut-off value may be preferable to use. A justification for our presentation of findings in relation to both cut-offs is now provided (see page 7, paragraph 4). 

- Has the analytical approach applied been discussed with a statistician?

Yes. Author Ari Samaranayaka is a biostatistician who was responsible for conducting the analyses reported in the manuscript. 

Results:

- Given the existing interest in differences between serving and ex-service personnel, all analyses should take serving status into account.

Our current multivariable analyses adjust for factors with a previously demonstrated relationship with PTS among military personnel, that is, age, sex, service years, and deployment status.

References:

- Owens GP, Steger MF, Whitesell AA, Herrera CJ. Posttraumatic stress disorder, guilt, depression, and meaning in life among military veterans. Journal of Traumatic Stress. 2009;22(6):654-7.

- Xue C, Ge Y, Tang B, Liu Y, Kang P, Wang M, Zhang L. A meta-analysis of risk factors for combat-related PTSD among military personnel and veterans. PloS One;10(3):e0120270.

Table 1:

- Need to report the overall numbers and %'s for each characteristic

The number for each overall characteristic is the same as that reported in the column heading. 

- Why are row and not column %'s reported?

Column percentages describe the distribution of each PCL-M group across categories of a characteristic. Row percentages compare the presence of PTS within each category of a characteristic. Which of these is more important (or easier to interpret) is dependent on personal preference. We used row percentages because they compare PTS within groups, which is consistent with what our main analysis method (i.e., logistic regression).

Table 3:

- Need to include numbers and %'s for each characteristic by PCL score (including those in the baseline category)

- A number of variables appear in this table which should be included in Tables 1 or 2, for example, hazardous drinking and trauma exposure

Numbers and percentages for each characteristic by PCL score are now provided in Table 1, including for categorical exposures (hazardous drinking and trauma exposure). 

Discussion:

- How do the prevalence estimates reported within the military population compare with the general population of New Zealand?

The prevalence estimates among the military personnel in our sample are significantly higher than those detected among the general New Zealand population. This information is now reported in the discussion (see page 18, paragraph 1). 

Conclusion:

- Please justify the statement regarding longitudinal data and how that will help identify those most at risk of PTS

Longitudinal research can be conducted to detect developments or changes in the characteristics of a target population over time. As a result, this type of research can be used to establish sequences of events (for example, whether changes in sleep precede changes in PTS). The utility of future longitudinal research has now been clarified in the discussion (see page 23, paragraph 1). 

Data Availability:

- The authors state that the full data won't be made available and that some restrictions will apply. Could they clarify what will be made available and what restrictions will be in place?

Our ethics approval does not allow us to share person level data, but summary data can be made available upon request to the corresponding author.

Editorial Requests:

Abstract

Interesting that this (PCL-M ≥30) is not much higher than other trauma population samples - worth discussing in the body of the paper?

Prior research has found that while substantial differences in PTS exist in response to different traumatic events, the weighted prevalence of PTS in response to war related trauma (3.5%) is similar to the weighted prevalence of PTS associated with random traumatic events (4%). This may explain why the prevalence of PTS symptomology found in our study is similar to that documented in other trauma population samples. The discussion has been updated to reflect this (see page 17, paragraph 1).

Exposure to trauma - Isn't this obvious that it would be associated? Could be better to focus on other novel findings in the abstract?

The sentence in the abstract reporting on the association between exposure to trauma and PTSD scores has now been removed in order to focus specifically on the novel findings of the study. 

Relationship between sleep and PTSD scores - Likely to be bi-directionally related to PTSD, though, given the prevalence and significance of sleep-related symptoms/problems in PTSD

We agree with the editor that the relationship between sleep and PTSD scores is likely to be bidirectional. This bidirectional relationship is now given attention in the discussion section of the manuscript (see page 20, paragraph 2). 

Introduction:

Please report the actual prevalence from these studies.

The actual prevalence of PTS found in the international studies described is now reported (see page 3, paragraph 3).

Fix wording/punctuation here.

The wording of this sentence has now been modified (see page 4, paragraph 1). 

Is there a reason to believe these would differ from other veteran samples?

There is reason to believe that risk and protective factors associated with PTS in this sample could differ from those identified in other populations because of the significant contextual variation that exists across countries. For example, the United States has a Veterans Health Administration that provides comprehensive and integrated healthcare specifically tailored to meet the unique needs of military personnel whereas such a system does not exist in New Zealand. This is now noted in the introduction (see page 4, paragraph 3). 

Prevalence of PTS - Not really likely to be achieved given the sampling techniques used (i.e., they're not likely to be representative of the target population).

We agree that the prevalence of PTS for all military personnel cannot be determined from our study due to the potential for sampling bias. We have modified this aim to read: ‘to determine the prevalence of PTS in a large sample of New Zealand military personnel’ (see page 5, paragraph 1). 

Methods:

Please explicitly state the inclusion and exclusion criteria.

All military personnel residing in New Zealand were eligible to participate in the study. With respect to exclusion criteria, individuals who had not served in the Armed Forces were excluded, such as members of the national police force and Fire and Emergency New Zealand staff and firefighters (see page 5, paragraph 2).

This is not a sufficient description of how biases were minimised. Please provide details on how the study was described to participants, what benefit they received from participating (if any) etc.

The methods section has now been updated to include more detail with respect to study procedures and the benefits associated with participation (see page 6, paragraph 2). 

How was the study described in the advertisements? It is likely to result in responder bias if it references PTSD prevalence etc.

A description of the content in study advertisements is now provided (see page 6, paragraph 2). It has also been made explicit that PTS was not mentioned in these advertisements, minimising the likelihood of responder bias. 

How did the authors choose six variables? Was this also a sufficient rationale for identifying the "prevalence" of PTS?

We have endeavoured to make it clear that 600 participants would be necessary to identify significant associations between risk and protective factors and PTS in our sample. These variables were chosen by reviewing potentially modifiable factors that had been identified in earlier studies (those of which are described in the introduction of our manuscript). This information has now been included in the analyses section of the manuscript (see page 10, paragraph 2). As the Editor has identified, it is not possible to determine the prevalence of PTS for all New Zealand military personnel as a consequence of our study design. We have addressed this by re-wording our objectives (see page 4, paragraph 3; page 5, paragraph 1).

PCL-M - Please state that this is the version of the tool that measures symptoms consistent with the DSM-IV diagnostic criteria.

It is now stated that the PCL-M is a self-report measure reflecting DSM-IV symptoms of PTS (see page 7, paragraph 4). 

Analyses:

RE: STROBE points 12d and 12e: sensitivity analyses should compare the estimates with respect to the sampling strategies (i.e., compare those from the whole sample with those recruited via the email advertisement or other advertisements). If the recruitment source was not recorded, the lack of sensitivity tests must be included in the discussion of limitations.

If these data were recorded you could use the suest command in stata to do these sensitivity tests.

Although we advertised the study in multiple ways, we cannot identify which participants responded to different types of advertisements as this information was not collected. Therefore, sensitivity tests were not conducted which is a limitation associated with the study. This is now noted in the discussion of limitations as recommended by the editor (see page 22, paragraph 2). We did collect information on whether participants completed an online or paper version of the questionnaire. Only a small number of participants (~4%) used a paper version and therefore we did not carry out a sensitivity analysis with respect to this.

Backward elimination process - Was this done manually?

The stepwise procedure was performed with Stata software. A manual elimination process would have produced the same results. 

Complete case analyses - Please justify this decision. Is it because the total number of cases with missing data was small? Why not use multiple imputation?

Biostatisticians do not consider multiple imputation (MI) to be an approach that can be used indiscriminately. The success of this approach is dependent on a number of assumptions, including the “missing at random” assumption, the ability of non-missing variables to predict missing variables, etc. Complete case analysis (i.e. the simplest approach to missing data) is appropriate if the amount of missing data is small. We had a moderate amount of missing data (approximately 15%). Therefore, there is some potential for bias associated with our complete case analysis, which is now acknowledged in the limitations section of the discussion (see page 22, paragraph 2). 

Results:

The letter to the editor references a "good response rate", but how was this determined given the recruitment strategies? Some effort to compare the demographic characteristics of the included sample with the total NZ military population would be useful to understand the representativeness of the cohort.

Information on the potential response rate has been updated in the discussion section of the manuscript (see page 21, paragraph 3). We have attempted to describe the distribution of the currently serving New Zealand military population to the best of our ability with the information that we have available. We have also acknowledged that we cannot compare the characteristics of retired military personnel in our sample with those of all retired military personnel in New Zealand as the number of individuals that make up this population is currently unknown (see page 21, paragraph 2). 

Table 2 could more succinctly be reported in text.

The results presented in Table 2 are now reported in text (see page 12, paragraph 2).

Tables 3 and 4 - Report absolute risk and the actual N, % with the outcome from participants included in each analysis. Why do a logistic regression? Did you consider ordinal regression where participants were classified as 0 = no clinically elevated symptoms, 1 = clinically elevated symptoms (30-44) and clinically significant symptoms (45+).

Table 3 presents univariate results and now includes the actual N for each categorical exposure variable as well as absolute risks. Table 4 now reports the actual N associated with categorical exposures. Percentages are not included in the tables as these can be calculated using the Ns provided to suit the preferences of individual readers (i.e. for row or column percentages). 

Although the use of ordinal logistic regression is attractive because it requires only one model as opposed to two logistic models, it can be used only under the assumption of proportionality. Unfortunately our data did not adhere to this assumption. 

Male sex finding - Interesting given that women typically have higher risk. Do you think that this could be an artefact of the sample, and that a higher proportion of men were exposed to trauma?

A 2014 NZDF report noted that only 6% of officers in combat/operations were women. Therefore, greater exposure to combat-related trauma among male military personnel in New Zealand may explain why they reported more PTS symptoms than female personnel. This information has been added to the discussion of the manuscript (see page 19, paragraph 3). 

Years in service finding - Discuss the potential that people with significant PTSD symptoms have shorter duration of service, and it is not the duration of service that plays a causal role in PTSS severity.

The discussion currently notes that the association between a greater number of service years and reduced odds of PTS may be due to individuals with PTS leaving military service earlier. A sentence has been added to emphasise that it is not the duration of service that plays a causal role in PTS severity (see page 19, paragraph 4).

Discussion:

Generally you should place the limitations and generalisability towards the end of the discussion - perhaps noting some of the key factors early (e.g., that the study can only provide a rough estimate of the prevalence of PTS given the sampling methods and response rate).

The description of limitations and generalisability of findings is now placed towards the end of the discussion, with only key factors mentioned by the editor identified at the beginning (see page 18, paragraph 1). 

Small response rate compared to other studies - This seems to be pretty small compared to other cohorts - discuss?

The response rate to this study was low relative to other studies conducted with military personnel which is likely attributable to the recruitment method used. In contrast to studies that recruited directly from veteran-specific treatment programs, we did not approach participants directly. Other studies have targeted pre-defined populations (e.g. US Gulf veterans) for which information on the approximate number of personnel is available, in addition to opportunities for direct contact. It is possible that New Zealand military personnel were not exposed to study advertisements and were therefore unaware that the study was taking place; this is particularly true of retired military personnel who were not emailed about the study. 

This information, along with appropriate references, is now provided in the discussion (see page 22, paragraph 1).

Comparison with UK military personnel - Explicitly name the variations here?

In contrast to our study, participants in the UK study were approached directly through an invitation pack, participated in clinical telephone interviews, and received a cheque or supermarket voucher to compensate them for their time (see page 18, paragraph 1).

Further discuss why you think males had higher risk.

This is now further discussed (see page 19, paragraph 3).

Please discuss sleep with respect to mechanisms of PTSD and sleep-related PTSD symptoms. Emphasise the correlational nature of the association, and that a causal relationship cannot be assumed.

We have now described the bidirectional relationship between PTSD and sleep problems in the discussion of the manuscript (see page 20, paragraph 2). We have also acknowledged that sleep related problems are a common symptom of PTS and emphasised the correlational nature of the association found in our study. 

Given that the trauma exposure could be from within military service or outside of service it would be good to see these types of trauma exposures were analysed separately.

The different types of trauma exposures experienced by participants was examined. Information on the number of participants who had served in a war zone and experienced childhood physical and sexual abuse is presented in the text (page 12, paragraph 3). Information on the proportion of participants experiencing all traumatic exposures is now also provided in the appendix. 

We would like to thank the reviewers and academic editor once more for their comments. We hope that with the specified changes the manuscript is now acceptable for publication.

---

## [Decision Letter · Decision Letter 1]

10 Feb 2020

PONE-D-19-23666R1

Risk and protective factors for post-traumatic stress among New Zealand military personnel: a cross sectional study

PLOS ONE

Dear Dr. McBride,

Thank you for submitting your manuscript to PLOS ONE. After careful consideration, we feel that it has merit but does not fully meet PLOS ONE’s publication criteria as it currently stands. Therefore, we invite you to submit a revised version of the manuscript that addresses the points raised during the review process.

To address the comments raised by Reviewer 2, please do add a column into Table 1 that shows the number (%) of people in the total sample belonging to each of the variable levels, and consider adding the variable "serving status" into Table 2 to demonstrate whether this factor is related to PTSD symptoms. If you choose not to include it in the multivariable analysis (e.g., due to multi-collinearity) then please explain that in your response.

We would appreciate receiving your revised manuscript by Mar 26 2020 11:59PM. To enhance the reproducibility of your results, we recommend that if applicable you deposit your laboratory protocols in protocols.io, where a protocol can be assigned its own identifier (DOI) such that it can be cited independently in the future. For instructions see: http://journals.plos.org/plosone/s/submission-guidelines#loc-laboratory-protocols

We look forward to receiving your revised manuscript.

Kind regards,

Melita J. Giummarra

Academic Editor

PLOS ONE

Reviewers' comments:

Reviewer's Responses to Questions

**Comments to the Author**

1. If the authors have adequately addressed your comments raised in a previous round of review and you feel that this manuscript is now acceptable for publication, you may indicate that here to bypass the “Comments to the Author” section, enter your conflict of interest statement in the “Confidential to Editor” section, and submit your "Accept" recommendation.

Reviewer #1: All comments have been addressed

Reviewer #2: (No Response)

2. Is the manuscript technically sound, and do the data support the conclusions?

Reviewer #1: Yes

Reviewer #2: Yes

3. Has the statistical analysis been performed appropriately and rigorously? 

Reviewer #1: Yes

Reviewer #2: Yes

4. Have the authors made all data underlying the findings in their manuscript fully available?

Reviewer #1: No

Reviewer #2: No

5. Is the manuscript presented in an intelligible fashion and written in standard English?

Reviewer #1: Yes

Reviewer #2: Yes

6. Review Comments to the Author

Reviewer #1: Comments have been addressed.

I do not think that the original or underlying datasets have been made available. So clarification regarding the methods of data access may be needed in line with the PLOS One policy.

The standard of English is good.

Reviewer #2: Thank for revising the manuscript based on the reviewers and editorial comments. The paper has been greatly improved. However, there are a couple of outstanding comments that require further consideration. I have detailed these below:

- An overall description of the study population is still absent. It is usual to have a "Table 1" which describes the characteristics of those included, an additional column could be added to the current Table 1.

- Given the existing interest in differences between serving and ex-service personnel, all analyses should take serving status into account. The authors state that "Our current multivariable analyses adjust for factors with a previously demonstrated relationship with PTS among military personnel, that is, age, sex, service years, and deployment status." But serving status has also been previously associated with PTS and thus should be taken into account in the analyses.

- Please ensure the references cited are up to date. The UK references used are old and more recent (and relevant) references are available. This is essential for the statements made regarding comparisons with UK studies.

7. PLOS authors have the option to publish the peer review history of their article (what does this mean?). If published, this will include your full peer review and any attached files.

Reviewer #1: Yes: Dr Michael Waller

Reviewer #2: No

---

## [Author Response · Author response to Decision Letter 1]

2 Mar 2020

We would like to thank the reviewers and academic editor for their considered comments on our manuscript. These comments are addressed below. Changes to the manuscript have been made using track changes.

Reviewer Two Comments:

An overall description of the study population is still absent. It is usual to have a "Table 1" which describes the characteristics of those included, an additional column could be added to the current Table 1.

Thank you for this recommendation. Characteristics of the total sample are now presented in an additional column in Table 1.

Given the existing interest in differences between serving and ex-service personnel, all analyses should take serving status into account. The authors state that "Our current multivariable analyses adjust for factors with a previously demonstrated relationship with PTS among military personnel, that is, age, sex, service years, and deployment status." But serving status has also been previously associated with PTS and thus should be taken into account in the analyses.

We thank the reviewer for alerting us to recent research demonstrating relationships between serving status and PTS. This research suggests that deployment and related stressors (such as having a combat role) may have different associations with PTS among currently serving and ex-serving personnel (Stevelink et al., 2018). However, the research has focused specifically on personnel who have deployed and is therefore based on participants who have been exposed to the stressors of deployment at some point in time (which may include combat exposure, discharging a weapon, witnessing someone being wounded or killed, and severe trauma). Our study was interested in examining predictive factors among all military personnel in New Zealand, including those who had never deployed. 

Stevelink SA, Jones M, Hull L, Pernet D, MacCrimmon S, Goodwin L, MacManus D, Murphy D, Jones N, Greenberg N, Rona RJ. Mental health outcomes at the end of the British involvement in the Iraq and Afghanistan conflicts: a cohort study. The British Journal of Psychiatry. 2018;213(6):690-7.

Please ensure the references cited are up to date. The UK references used are old and more recent (and relevant) references are available. This is essential for the statements made regarding comparisons with UK studies.

Thank you for bringing this to our attention. A more recent reference for a widely cited study with a larger and more representative sample of UK military personnel has now been used throughout the manuscript. 

Editorial Requests:

To address the comments raised by Reviewer 2, please do add a column into Table 1 that shows the number (%) of people in the total sample belonging to each of the variable levels.

Please see response to Reviewer 2, comment one. 

Consider adding the variable "serving status" into Table 2 to demonstrate whether this factor is related to PTSD symptoms. If you choose not to include it in the multivariable analysis (e.g., due to multi-collinearity) then please explain that in your response.

Please see response to Reviewer 2, comment two.

We would like to thank the reviewers and academic editor once more for their comments. We hope that with the specified changes the manuscript is now acceptable for publication.

---

## [Editor Report · Decision Letter 2]

25 Mar 2020

Risk and protective factors for post-traumatic stress among New Zealand military personnel: a cross sectional study

PONE-D-19-23666R2

Dear Dr. McBride,

We are pleased to inform you that your manuscript has been judged scientifically suitable for publication and will be formally accepted for publication once it complies with all outstanding technical requirements.

With kind regards,

Melita J. Giummarra

Academic Editor

PLOS ONE
---

## [Editor Report · Acceptance letter]

1 Apr 2020

PONE-D-19-23666R2 

Risk and protective factors for post-traumatic stress among New Zealand military personnel: a cross sectional study 

Dear Dr. McBride:

I am pleased to inform you that your manuscript has been deemed suitable for publication in PLOS ONE. Congratulations! Your manuscript is now with our production department. 

With kind regards,

on behalf of

Dr. Melita J. Giummarra 

Academic Editor

PLOS ONE